# Drosophila fabp is required for light-dependent Rhodopsin-1 clearance and photoreceptor survival

**Huai-Wei Huang**, **Hyung Don Ryoo***

Department of Cell Biology NYU Grossman School of Medicine New York, New York, United States of America

* hyungdon.ryoo@nyumc.org

## Abstract

Rhodopsins are light-detecting proteins coupled with retinal chromophores essential for visual function. Coincidentally, dysfunctional Rhodopsin homeostasis underlies retinal degeneration in humans and model organisms. Drosophila $ninaE^{G69D}$ mutant is one such example, where the encoded Rh1 protein imposes endoplasmic reticulum (ER) stress and causes light-dependent retinal degeneration. The underlying reason for such light-dependency remains unknown. Here, we report that Drosophila fatty acid binding protein (fabp) is a gene induced in $ninaE^{G69D}/+$ photoreceptors, and regulates light-dependent Rhodopsin-1 (Rh1) protein clearance and photoreceptor survival. Specifically, our photoreceptor-specific gene expression profiling study in $ninaE^{G69D}/+$ flies revealed increased expression of fabp together with other genes that control light-dependent Rh1 protein degradation. fabp induction in $ninaE^{G69D}$ photoreceptors required vitamin A and its transporter genes. In flies reared under light, loss of fabp caused an accumulation of Rh1 proteins in cytoplasmic vesicles. The increase in Rh1 levels under these conditions was dependent on Arrestin2 that mediates feedback inhibition of light-activated Rh1. fabp mutants exhibited light-dependent retinal degeneration, a phenotype also found in other mutants that block light-induced Rh1 degradation. These observations reveal a previously unrecognized link between light-dependent Rh1 proteostasis and the ER-stress imposing $ninaE^{G69D}$ mutant that cause retinal degeneration.

## Author summary

Rhodopsins are light-detecting proteins that use retinoids as chromophore co-factors. Rhodopsins are tighly regulated in photoreceptors, as dysfunctional Rhodopsins cause photoreceptor degeneration. The precise mechanisms by which photoreceptors regulate Rhodopsin homeostasis remains unclear. Here, we report that Drosophila fatty acid binding protein (fabp) is a gene required for Rhodopsin-1 (Rh1) protein homeostasis and photoreceptor survival. Specifically, we found that fabp is among the genes induced by an endoplasmic reticulum (ER) stress-imposing Rhodopsin-1 (Rh1) mutant, $ninaE^{G69D}$, which serves as a Drosophila model for Retinitis Pigmentosa. We further found that fabp

**Data Availability Statement:** All RNA-sequencing raw data are available from NIH GEO (accession number GSE185134). All other relevant data are within the manuscript and its Supporting information files.

**Funding:** This work was supported by the National Institutes of Health grant R01 EY020866 to H.D.R. (https://www.nei.nih.gov). The funders had no role in study design, data collection and analysis, decision to publish, or preparation of the manuscript.

**Competing interests:** The authors have declared that no competing interests exist.

induction in $ninaE^{G69D}$ photoreceptors required vitamin A and its transporter genes. *fabp* was required in photoreceptors to help degrade light-activated Rh1. In the absence of *fabp*, Rh1 accumulated in cytoplasmic vesicles in a light-dependent manner, and exhibited light-dependent retinal degeneration. These observations indicate that *fabp* is required for light-induced Rh1 degradation and photoreceptor survival.

## Introduction

Rhodopsins are G-protein coupled proteins associated with retinal chromophores that detect light and initiate signal transduction [1]. As in mammals, *Drosophila* has multiple Rhodopsins, including *ninaE* (*neither inactivation nor afterpotential*) that encodes the Rhodopsin-1 (Rh1) protein expressed in R1 to R6 photoreceptors [2–4]. Functional Rh1 is covalently attached to the 11-cis-3-hydroxyretinal chromophore, which is derived from dietary vitamin A [5–7]. *ninaE* loss of function results in an impairment of visual function [4,8].

Abnormal Rh1 protein homeostasis is a frequent cause of retinal degeneration. One class is caused by a group of *ninaE* missense mutations that dominantly cause progressive age-related retinal degeneration [9,10]. These alleles are analogous to human Rhodopsin mutations that underlie age-related retinal degeneration in Autosomal Dominant Retinitis Pigmentosa (ADRP) patients [11,12]. Using the *Drosophila ninaE^{G69D}* allele as a model, we previously established that these mutations impose stress in the endoplasmic reticulum (ER), which contributes to retinal degeneration [13,14]. The human Rhodopsin allele that is most frequently found associated with ADRP, the P23H mutant, similarly imposes stress in the ER of mammalian cells [15]. Notably, the retinal degeneration phenotype of *ninaE^{G69D}*/+ flies is light-dependent [9,10], but the underlying reason remains unknown. There is as yet no evidence that light exposure affects the degree of ER stress imposed by the mutant protein.

Cellular mechanisms that regulate Rhodopsin protein levels affect retinal degeneration. Flies bearing one copy of the *ninaE^{G69D}* allele have total Rh1 protein levels reduced by more than half, indicating that both the mutant and the wild type Rhodopsin-1 proteins undergo degradation in these flies [9,10]. Three ubiquitin ligases that specialize in the degradation of misfolded endoplasmic reticulum (ER) proteins mediate the degradation of Rh1 in *ninaE^{G69D}* flies [16]. Overexpression of these ubiquitin ligases can delay the onset of retinal degeneration in *Drosophila ninaE^{G69D}* flies, suggesting that excessive ER stress imposed by mutant Rh1 is a contributing factor to retinal degeneration [14,16].

Functional wild type Rh1 proteins also undergo degradation after being activated by light. Specifically, this occurs after light-activated Rh1, also referred to as metarhodopsin (M), engages with Arrestin that mediates feedback inhibition [17]. Rh1 forms a stable complex with Arrestin and together undergo endocytosis [18–21]. The photoreceptors need to degrade those endocytosed Rh1, as excessive Rh1 accumulation in the endosome/lysosome can cause light-dependent retinal degeneration [18,19,22–26]. These aspects appear to be conserved across phyla, as the human Rhodopsin mutants that exhibit high affinities for Arrestin display endosomal abnormalities and are associated with severe forms of ADRP [27,28].

Retinoids are among the molecules that affect Rh1 protein levels. Deprivation of vitamin A, which serves as a precursor for the retinal chromophore, causes a reduction in overall Rh1 levels [29–33]. Such an effect is largely attributed to the importance of chromophores in Rh1 protein maturation. Aside from its role as a Rhodopsin cofactor, retinoids have other functions in vertebrates, including the regulation of gene expression through nuclear hormone receptors [34]. For these alternative retinoid functions, Cellular Retinoic Acid Binding Protein -I and -II

(CRABP-1, -II) bind to the lipophilic retinoic acids (RA) and deliver them to specific subcellular sites [35,36]. In addition to mediating the RA signaling response, CRABP-II itself is induced by RA signaling [37]. Whether similar retinoid binding proteins play important biological roles in *Drosophila* remain unclear [38,39]. Intriguingly, a recent study reported that vitamin A deficiency affects the expression of several genes in *Drosophila*, including *Arrestin1* and *2* [40].

Here, we report that Rh1 protein levels are regulated by the *Drosophila* CRABP homolog, *fatty acid binding protein* (*fabp*). Specifically, we found that *fabp* is among the genes induced in *ninaE*$^{G69D}$/+ photoreceptors. Loss of *fabp* enhances total Rh1 levels in *ninaE wild type* and *G69D* mutant backgrounds dependent on light and *Arrestin2*. Moreover, loss of *fabp* causes light-dependent retinal degeneration. Our results reveal a link between the ER-stress imposing *ninaE*$^{G69D}$ mutant and light-activated Rh1 degradation through endosomes. This link provides clues regarding the light-dependent nature of the *ninaE*$^{G69D}$/+ retinal degeneration phenotype.

## Results

### Photoreceptor-specific gene expression profiling shows *fabp* induction in *ninaE*$^{G69D}$ eyes

To better understand how photoreceptors respond to stress imposed by the *ninaE*$^{G69D}$ allele, we performed a photoreceptor-specific gene expression profiling analysis. We specifically employed a previously described approach in which the expression of the nuclear envelope-localized *EGFP*::*Msp300*$^{KASH}$ is driven in specific cell-types through the Gal4/UAS system to isolate the EGFP-labeled nuclei for RNA-seq analysis [41,42]. We used the *Rh1-Gal4* driver to isolate *ninaE* expressing R1 to R6 photoreceptor nuclei from the adult fly ommatidia (Fig 1A). Microscopy imaging confirmed that anti-EGFP beads enriched the EGFP::Msp300$^{KASH}$-tagged nuclei (Fig 1B and 1C). RNA-seq was performed with nuclei isolated from *ninaE wild type* and *ninaE*$^{G69D}$/+ photoreceptors. The sequencing results have been deposited to NIH GEO (GSE185134).

Differential gene expression analysis showed 182 genes whose expression changed with adjusted p values below 0.01 (S1 Table). Among the most highly induced genes was *gstD1* (Fig 1D), which was also identified as an ER stress-inducible gene in a separate study performed with larval imaginal discs [43]. *cnx99A* [44], which encodes an ER chaperone essential for Rh1 maturation, was also induced significantly (Fig 1D). Another ER chaperone, Hsc70-3 (also known as BiP), was induced at a more moderate level (log$_2$ Fold Change = 0.355, adjusted p value = 0.095; see S1 Table). These observations are consistent with the previous report that *ninaE*$^{G69D}$ imposes ER stress in photoreceptors [13].

Also, notable from the differential gene expression analysis was the induction of genes that could affect Rh1 levels. *ninaE* was itself induced in *ninaE*$^{G69D}$ samples (Fig 1D). Since *ninaE*$^{G69D}$/+ flies have very low Rh1 levels [9,10], we speculate that increases in *ninaE* transcription may be part of a feedback homeostatic response. Also induced were genes *Arrestin1* (*Arr1*), *Arrestin2* (*Arr2*) and *culd* (Fig 1D), which promote the endocytosis of light-activated Rh1 in photoreceptors [17–21,45].

Among the *ninaE*$^{G69D}$-induced genes was *fabp* (Fig 1D), which encodes a protein homologous to human CRABP-1, -II and FABP5 (Fig 1E). We validated the induction of *fabp* mRNA and protein in *ninaE*$^{G69D}$/+ through q-RT-PCR and western blots (Fig 1F–1H). The human homologs of *fabp* are known to bind all trans RA with high affinity [35,36,46,47]. Notably, CRABP-II is one of the well-characterized RA inducible genes in mammalian cells [37].

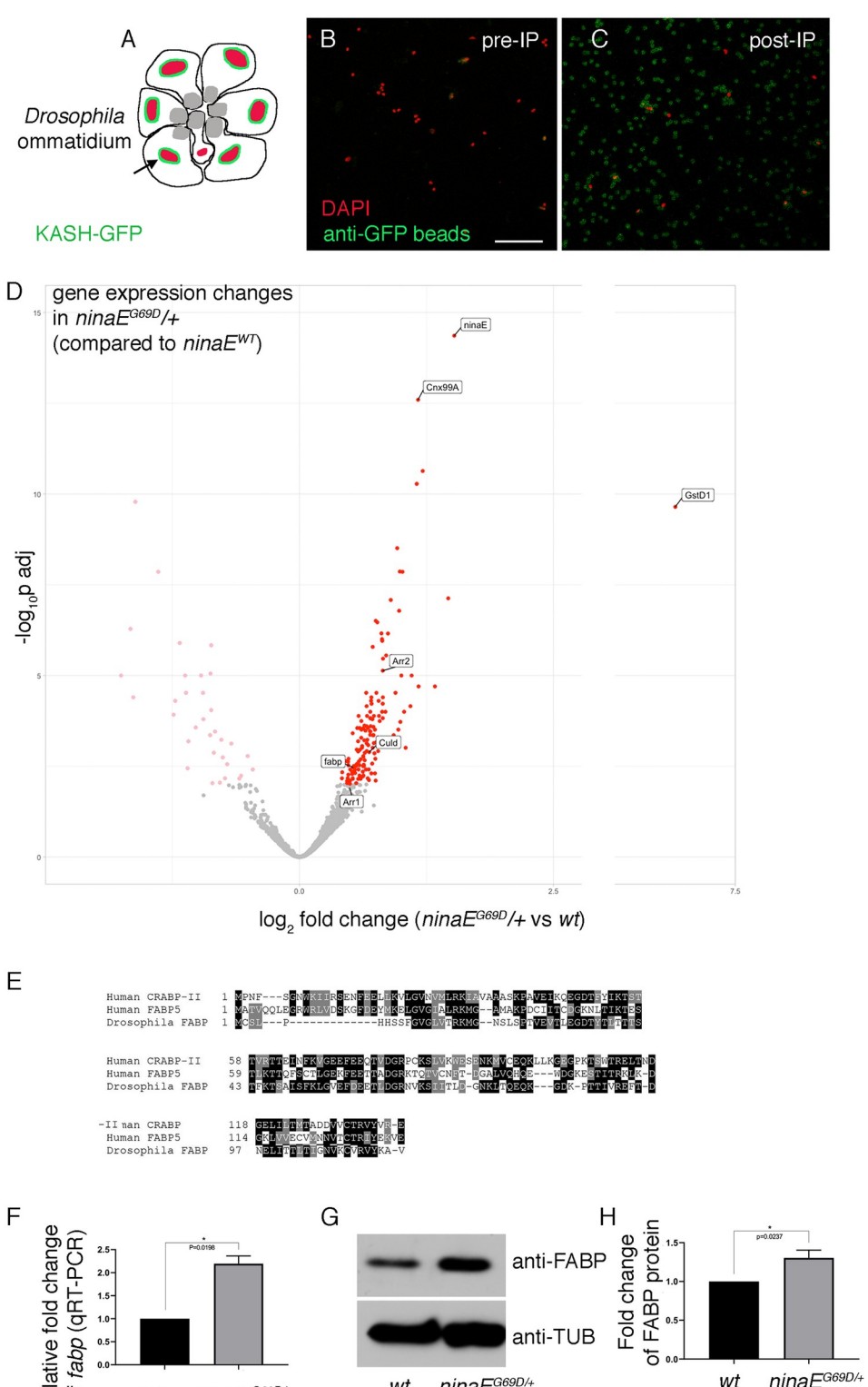

**Fig 1. Photoreceptor-specific gene expression profiling in *ninaE*^G69D^ eyes.** (A) A schematic diagram of the *Drosophila* ommatidium. Shown are seven photoreceptor cells, R1 to R7. KASH-GFP (green) coats the outer membranes of R1 to R6 nuclei (red). Rh1 localizes to the apical membrane structure known as rhabdomeres (gray). (B, C) Purification of photoreceptor nuclei tagged with EGFP-Msp300^KASH (KASH-GFP). *Rh1-Gal4* was used to express *uas-KASH-GFP* in R1 to R6 photoreceptors. Nuclei are labeled with DAPI (red) and the anti-GFP beads are in green.

Scale bar = 50 μm. (B) Before anti-GFP bead purification. (C) After purification, the DAPI labeled nuclei are associated with the beads (green). (D) Volcano plot of differential gene expression compared between *ninaE wild type* and *ninaE*<sup>G69D</sup>/+ photoreceptors. The y axis shows -log$_{10}$ (adjusted p value). The x axis represents log$_2$ fold change, with those whose expressions increase in *ninaE*<sup>G69D</sup>/+ on the right (adjusted p<0.01 are labeled as red dots). Log$_2$ fold change above 2.5 is not in scale. Genes with nonsignificant changes (adjusted p>0.01) are represented as gray dots. (E) Sequence comparison between *Drosophila* FABP, human CRABP-II and FABP5. (F) q-RT-PCR results of *fabp* from *ninaE wild type* (left) and *ninaE*<sup>G69D</sup>/+ fly heads. (G) Anti-FABP (top gel) and anti-β-Tubulin (bottom gel) western blots of *ninaE wild type* (left) and *ninaE*<sup>G69D</sup>/+ fly heads. (H) Quantification of normalized anti-FABP band intensities of the indicated genotypes. Error bars represent Standard Error (SE). t-test was used to assess statistical significance. * = p<0.05.

## *fabp* expression requires vitamin A and retinoids

To test if *Drosophila fabp* is also affected by retinoic acids (RA), we examined *fabp* mRNA levels through q-RT-PCR in *Drosophila* S2 culture cells treated with or without 10mM RA. We found that RA treated cells showed an increase in *fabp* transcripts after 60 minutes of RA exposure (Fig 2A and 2B). Consistently, FABP protein levels also increased after 3 hours of RA exposure (Fig 2C and 2D).

To examine if *fabp* expression in fly tissues is affected by the availability of vitamin A and its metabolites, we employed *ninaD* and *santa maria* mutants that have impaired transport of carotenoids. These mutants are devoid of retinoids in the retina, as evidenced by defective rhodopsin maturation and light detection [30,33]. We found that these mutants had reduced FABP protein as assessed through western blot of fly head extracts (Fig 2E). Consistently, the mutants also had reduced *fabp* mRNA levels as assessed through q-RT-PCR (Fig 2F).

These observations prompted us to examine if other *ninaE*<sup>G69D</sup>-inducible genes require *santa maria* and *ninaD* for proper expression. We focused on candidates known to be involved in Rh1 protein regulation. Among those tested, the mRNAs of *Arr1* and *Arr2* were found to be reduced in *ninaD* or *santa maria* mutant backgrounds (Fig 2G and 2H). These results are consistent with a recent study reporting a reduction of *Arr1* and *Arr2* expression in flies deprived of vitamin A in the diet [40]. However, not all genes involved in Rh1 homeostasis were affected in these mutants. For example, *fatty acid transport protein* (*fatp*) is a gene whose loss-of-function increases Rh1 protein levels [26]. *fatp* mRNA levels were affected neither in the mutant backgrounds of *ninaD* nor *santa maria* (Fig 2I). These results indicate that the expression of *fabp*, *Arr1*, *and Arr2* specifically require retinoid and carotenoid transporters, *ninaD* and *santa maria*.

## An *fabp* protein trap line shows carotenoid-dependent expression in the larval intestine

To independently validate the carotenoid-dependent expression of *fabp in vivo*, we utilized the *fabp* protein trap line *CA06960*. This P-element insertion line has a GFP with splice donor and acceptor sites, designed to make fusion proteins with the endogenous *fabp* coding sequence (Fig 3A). Anti-GFP western blot of fly extracts confirmed the expression of a GFP-fused protein in adult fly head extracts with the predicted size (Fig 3B).

In the third instar larva, the *fabp*<sup>CA06960</sup> line had GFP expression detectable in several regions of the intestine (Fig 3C and 3E). Such expression was abolished when the flies were reared in vitamin A deficient food (Fig 3D). Consistently, the expression of GFP was suppressed in the mutant backgrounds of *ninaD* and *santa maria* (Fig 3F and 3G). In adult flies, the GFP signal was most prominent in the female abdomen, which was reduced in the *ninaD* mutant background (Fig 3H). These results independently support the idea that *fabp* expression depends on carotenoids.

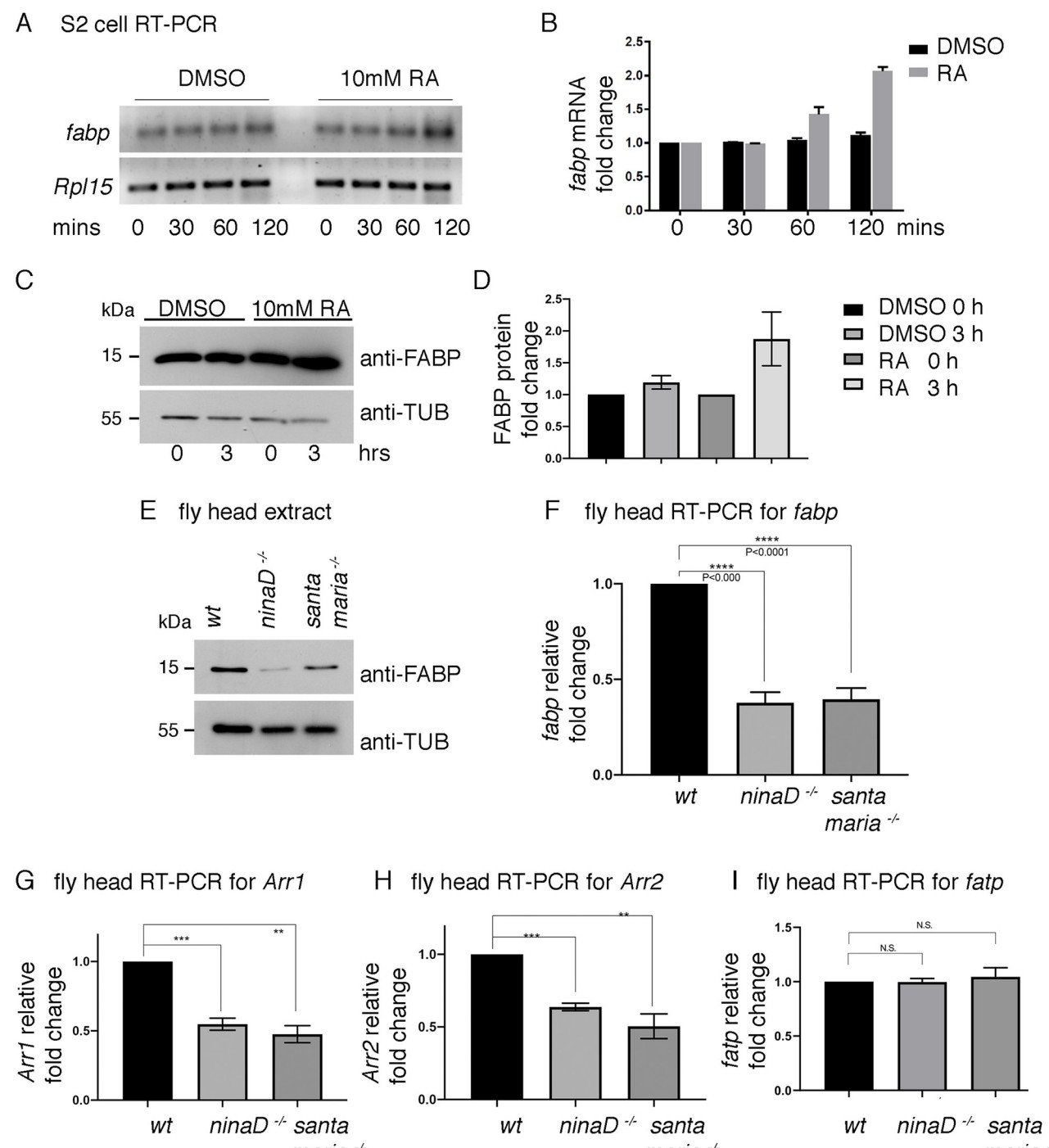

**Fig 2. *fabp* expression is regulated by carotenoids and retinoic acid.** (A) The time course of *fabp* mRNA induction as assessed through semi-quantitative RT-PCR for *fabp* (top gel) and the control *Rpl15* (bottom gel). Cultured *Drosophila* S2 cells were treated with either control DMSO (left 4 lanes) or with 10mM all-trans retinoic acid (RA, right 4 lanes) for the indicated periods of time. (B) q-RT-PCR of *fabp* from S2 cells treated with DMSO (black) or 10mM all-trans retinoic acid (grey bars) for the indicated period of time. The y axis shows fold induction as compared to the results from the DMSO controls. (C) Western blot of FABP (top gel) and β-Tubulin (bottom gel) from S2 cell extracts exposed to control DMSO (lanes 1, 2) and 10mM RA (lanes 3, 4) for the indicated time periods. (D) Quantification of relative FABP protein band intensities from (C). (E) Western blots for FABP (top) and Tubulin (bottom) from adult fly head extracts of the indicated genotypes. $w^{1118}$ flies were used as *wild type* controls. (F-I) q-RT-PCR-based assessment of indicated mRNAs from adult fly heads of the indicated genotypes. The levels of *fabp* (F), *Arr1* (G), *Arr2* (H), *fatp* (I) are shown. The y axis shows fold changes compared to results obtained from *wild type* control samples. In all q-RT-PCRs, *RpL15* q-RT-PCR results were used to normalize the levels of transcripts of interest. Error bars represent standard error (SE). Statistical significance was assessed through two tailed t-tests. ** = p<0.005, *** = p<0.0005, **** = p<0.0001.

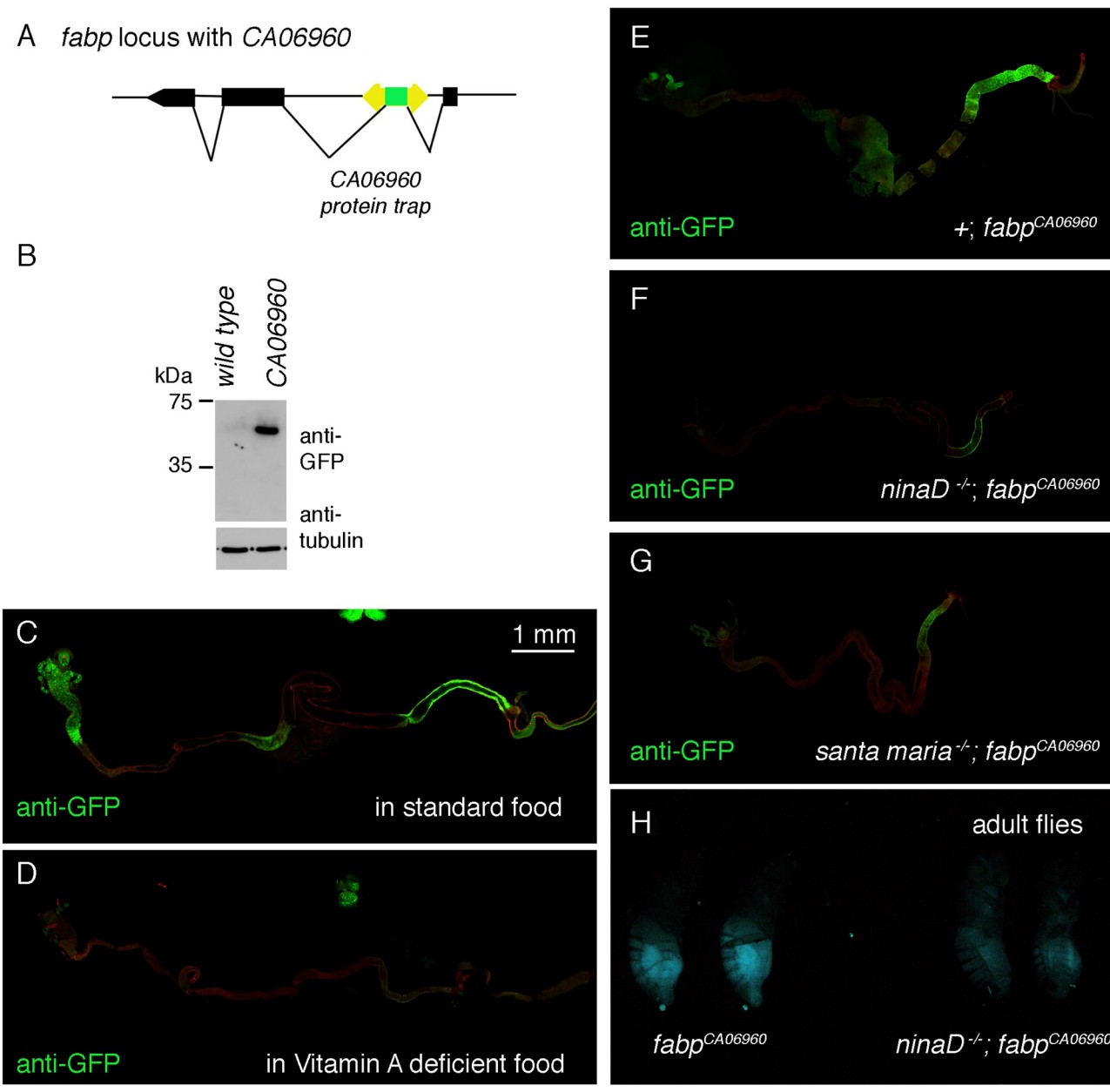

**Fig 3. The expression of an *fabp* GFP protein trap line requires vitamin A and its transporter genes.** (A) The structure of the *fabp* locus and the *CA06960* protein trap line. (B) Anti-GFP western of adult head extracts from control and the *CA06960* protein trap line. (C-G) Images of dissected *fabp*$^{CA06960}$ third instar larval intestines immuno-labeled with anti-GFP antibody (green). GFP signal is detected in distinct regions of the intestine in flies reared under standard conditions (C), which decreases in those reared with vitamin A deficient food (D). (E-G) GFP signal from flies reared under standard food in the control genotype (E), in the *ninaD*$^1$ mutants (F) and in the *santa maria*$^1$ mutant background (G). (H) GFP signal of *fabp*$^{CA06960}$ adult females in the control genetic background (left two flies) and in the *ninaD*$^1$ mutant background (right two flies). The scale bar in C is for images C-G.

## Loss of *fabp* increases Rh1 protein levels

*ninaE*$^{G69D}$/+ flies have drastically reduced Rh1 protein levels as compared to *ninaE wild type* flies. Using this property as a readout, we have been performing targeted RNAi screens to identify regulators of Rh1 degradation in *ninaE*$^{G69D}$/+ photoreceptors [48]. Specifically, we drove the expression of RNAi lines that target the genes of interest in the photoreceptors using

the Rh1-Gal4/UAS system (S1 Fig). Of interest for this study were potential genes involved in RA metabolism and signaling, including enzymes that convert vitamin A to retinoids (e.g. *ninaB* [49]), nuclear hormone receptors (e.g. knl and eg) and *fabp*. Most lines had no effect on Rh1 levels (S1 Fig). The negative results are consistent with the view that *Drosophila* doesn't have an RA signaling mechanism analogous to those in vertebrates. It is equally possible that the negative outcome is due to insufficient RNAi knockdown efficiency, or perhaps because those genes act outside the photoreceptor cells where the RNAi lines were expressed. Interestingly, an RNAi line that targeted *fabp* showed a reproducible effect of partially enhancing Rh1 levels as assessed through western blots of fly head extracts (S1 Fig).

To validate *fabp* RNAi results, we employed an *fabp* loss of function allele, *EY02678*, which has a P-element inserted near an exon-intron boundary (Fig 4A). This allele has strongly reduced FABP expression as assessed through western blot (Fig 4B). Rh1 transcript levels did not change significantly in the *fabp*$^{EY06747}$ mutants as assessed through q-RT-PCR (Fig 4C). By contrast, Rh1 protein levels increased in the *fabp*$^{EY02678}$ -/- background as compared to *fabp*+ controls, both in the *ninaE*$^{G69D}$/+ and *ninaE wild type* flies (Fig 4D and 4E), When we re-introduced *fabp* expression in *fabp* mutant flies using the eye specific *GMR-Gal4* driver, Rh1 protein levels were restored to those levels of *fabp wild type* controls (Fig 4D and 4E). These results indicate that *fabp* affects general Rh1 protein levels.

The very low levels of overall Rh1 detected in *ninaE*$^{G69D}$/+ fly heads indicates that both the wild type and the Rh1$^{G69D}$ proteins undergo degradation in this genetic background. To test if *fabp* mutants stabilize the *wild type* Rh1 protein in this genotype, we used a Rh1 transgenic line in which the Rh1 promoter drives the expression of a *wild type* Rh1 coding sequence tagged with an HSV epitope [50]. This HSV epitope was detected at high levels in control *ninaE wild type* background, but was detected at significantly lower levels in the *ninaE*$^{G69D}$/+ background, confirming the idea that the *G69D* allele destabilizes the *wild type* Rh1 protein. Such effect of *ninaE*$^{G69D}$ on Rh1$^{WT}$-HSV was reversed in *fabp*$^{EY02678}$ flies (Fig 4F and 4G). These results indicate that *wild type* Rh1 protein becomes stabilized in response to *fabp* loss.

## Regulation of Rh1 by *fabp* is dependent on light and *Arrestin2*

Since wild type Rh1 proteins are most notably degraded through light-dependent endocytosis [18–21], we examined whether *fabp* regulation of Rh1 was light-dependent. We found that *fabp* mutants showed higher Rh1 levels when the flies were reared under constant exposure of moderate light (1000 lux). Similar effects were observed when flies were exposed to blue light for three hours (S2 Fig). However, such effect was not seen in flies that were reared in dark (Fig 5A and 5B).

Light-dependent Rh1 endocytosis is initiated by Arrestins, Arr1 and Arr2. In photoreceptors, Arr2 is the major Arrestin involved in this negative feedback loop [17]. *Arr2*$^{3}$ mutant flies have total Rh1 protein levels similar to wild type controls [51]. To test if *Arr2* genetically interacts with *fabp*, we examined Rh1 levels in *Arr2*$^{3}$; *fabp*$^{EY06747}$ double mutants reared under constant light exposure (1000 lux). We found that increases in Rh1 caused by *fabp* loss was suppressed in *Arr2*$^{3}$; *fabp*$^{EY06747}$ fly heads (Fig 5C and 5D). Similar effects were seen when the flies were exposed to blue light for three hours (S2 Fig). Together with the finding that *fabp* mutants affect Rh1 levels specifically under light, these results indicate that *fabp* is involved in the light-activated Rh1 degradation pathway initiated by *Arr2*.

We further examined if *fabp* genetically interacts with other genes involved in light-dependent Rh1 degradation. Western blots from fly head extracts show increased Rh1 protein levels in *fabp*$^{EY06747}$ flies reared under light. Such increases are suppressed when transgenic *fabp* is expressed in that background using the *GMR-Gal4* driver (Fig 5E and 5F, compare lanes 1–3).

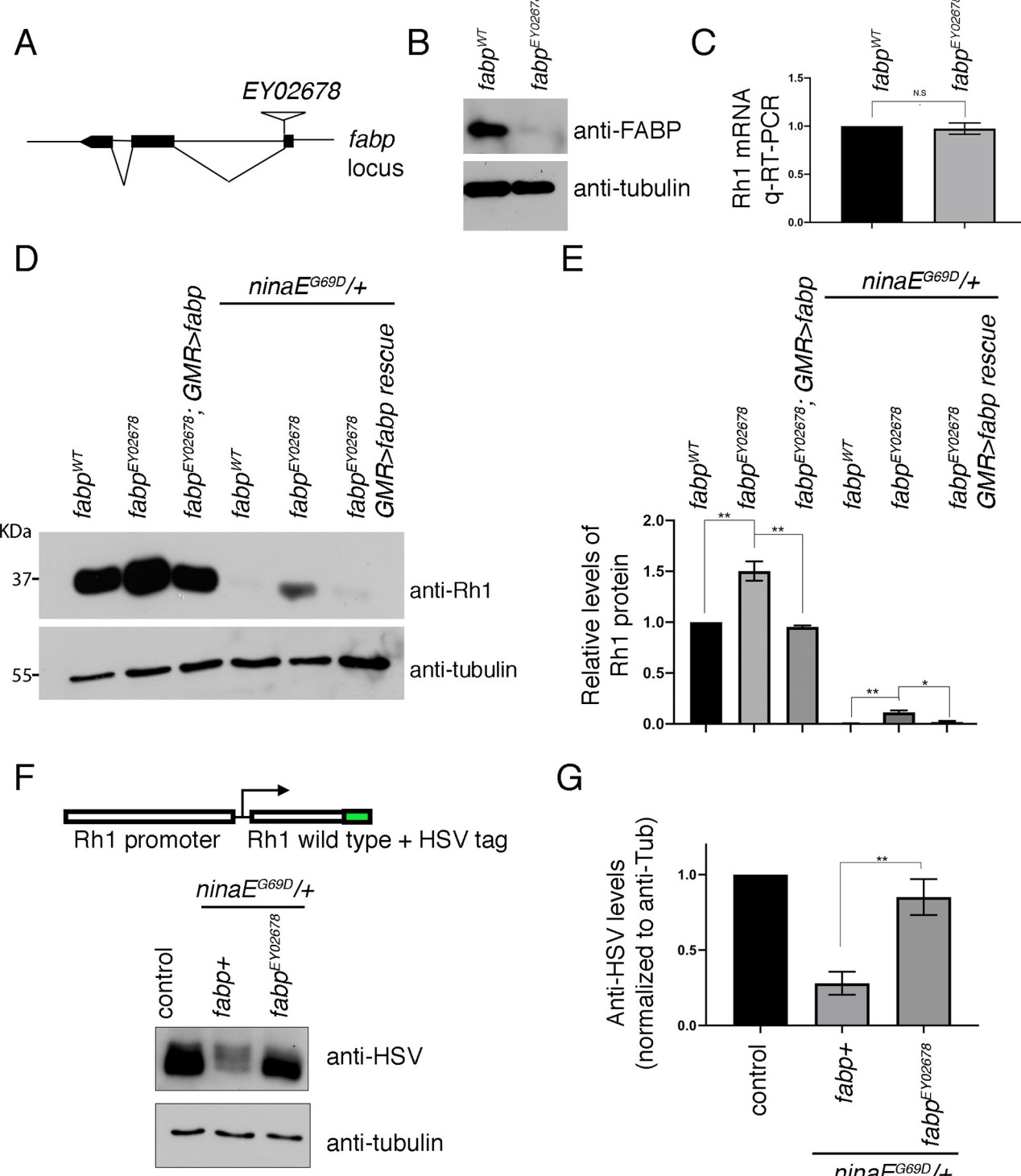

**Fig 4. Loss of *fabp* enhances Rhodopsin-1 protein levels.** (A) A schematic diagram of the *fabp* locus and the *EY02678* P-element insertion site. (B) Evidence that *fabp^EY02678* is a loss-of-function allele. Shown are western blots of FABP and β-Tubulin from adult fly head extracts of wild type or *fabp^EY02678* flies. (C) Rh1 mRNA q-RT-PCR from fly heads of *fabp wild type* control (left) and *fabp^EY02678* mutants (right). (D) Western blot of Rhodopsin-1 (Rh1) and β-Tubulin from fly heads extracts of the indicated genotypes: *fabp wild type* (lane 1), *fabp^EY02678* mutants (lane 2), *fabp^EY02678* mutants rescued with *GMR-Gal4* driven *uas-fabp* expression (lane 3). Lanes 4–6 show anti-Rh1 blots in the *ninaE^G69D*/+ background, with *fabp wild type* (lane 4), *fabp^EY0678* (lane 5), and *fabp^EY02678* mutants rescued with *GMR-Gal4/uas-fabp* (lane 6). (E) Quantification of relative Rh1 band intensities as normalized to β-Tubulin. (F) Western blot of HSV-tagged transgenic Rh1 from head extracts. The schematic diagram (top) shows that HSV-tag is

fused to the C-terminus of the Rh1 coding sequence, and this fusion protein is expressed through Rh1's promoter. Genotypes: *fabp* and *ninaE* wild type control (lane 1). *ninaE*$^{G69D}$/+ in the *fabp wild type* background (lane 2). *ninaE*$^{G69D}$/+ in the *fabp*$^{EY02678}$ background. (G) Quantification of relative Rh1 band intensities as normalized with β-Tubulin. Error bars represent standard error (SE). Statistical significance was assessed through two tailed t-tests. * = p<0.05, ** = p<0.005

We found no evidence of genetic interaction between *fabp* and *fatty acid transport protein* in this assay. Specifically, expressing *fatty acid transport protein* (*fatp*) under equivalent conditions did not reduce Rh1 levels in the *fabp* mutant eyes (Fig 5E, lane 4). Knockdown of *fatty acid transport protein* increased Rh1 levels (Fig 5E and 5F, lane 6), as had been reported previously [26]. Overexpressing *fabp* in that genetic background had no effect on the *fatty acid transport protein* RNAi phenotype.

Interestingly, we found that *fabp* genetically interacted with *Vps26*, a subunit of the retromer complex with established roles in Rh1 recycling from endosomes [24]. Specifically, *Vps26* expression reduced Rh1 levels in *fabp* mutants (Fig 5E and 5F, lane 5). Together with the results with *Arr2* mutants, these results indicate that *fabp* is involved in the regulation of light-dependent Rh1 endocytosis and degradation.

## Rh1 protein localization in *fabp* mutants

To examine the pattern of Rh1 distribution in photoreceptors, we performed anti-Rh1 immuno-labeling in the adult *Drosophila* retina. In control wild type flies, Rh1 is predominantly detected in the rhabdomeres of R1 to R6 photoreceptor cells organized in a trapezoidal pattern (Fig 6A–6E). In *fabp*$^{EY06747}$ -/- flies reared under light, however, there were additional anti-Rh1 signals in intracellular vesicles (Fig 6D).

Vesicular Rh1 signals reportedly appear in flies exposed to light, becoming even more prominent in mutants that have defects in Rh1 trafficking to the lysosome [23, 24]. We found that vesicular Rh1 patterns in *fabp*$^{EY06747}$ -/- eyes were also light-dependent, as extra-rhabdomeric anti-Rh1 signals mostly disappeared in flies raised under constant darkness (Fig 6E).

To gain further insights regarding Rh1 distribution in *fabp*$^{EY06747}$ -/- fly eyes, we performed double labeling experiments with *ninaE-GFP* transgenic flies. This transgene has GFP fused in frame with the Rh1 coding sequence to visualize Rh1 protein localization. When crossed into the *fabp*$^{EY06747}$ -/- background and reared under light, the GFP signal was detected in and outside of the rhabdomeres. Those signals outside the rhabdomeres showed partial overlap with the early endosome marker, Rab5 (Fig 6F), and with the late endosome/lysosome marker Rab7 (Fig 6G). There were additional GFP signals that did not overlap with Rab5 and Rab7. Partial overlap with these markers suggests that Rh1 localize to several different intracellular sites, including endosomes, in *fabp* mutants.

## *fabp* mutants show light-dependent retinal degeneration that is sensitive to Rh1 levels

To examine whether *fabp* mutants affect retinal degeneration, we reared wild type and *fabp*$^{EY06747}$ -/- flies under moderate light (see Materials and methods; Retinal degeneration assay), and performed transmission electron microscopy (TEM) imaging to visualize their ommatidia. Wild type control flies have repeating units of ommatidia, each showing seven rhabdomeres arranged in a trapezoidal pattern (Fig 7A). Such ommatidial arrangement was severely disrupted in *fabp*$^{EY06747}$ -/- flies at 27 days after eclosion. Some ommatidia had less than seven rhabdomeres per ommatidia, indicative of retinal degeneration (Fig 7B). There were numerous vacuoles in between the ommatidia, and the array of ommatidial units were generally distorted (S3 Fig).

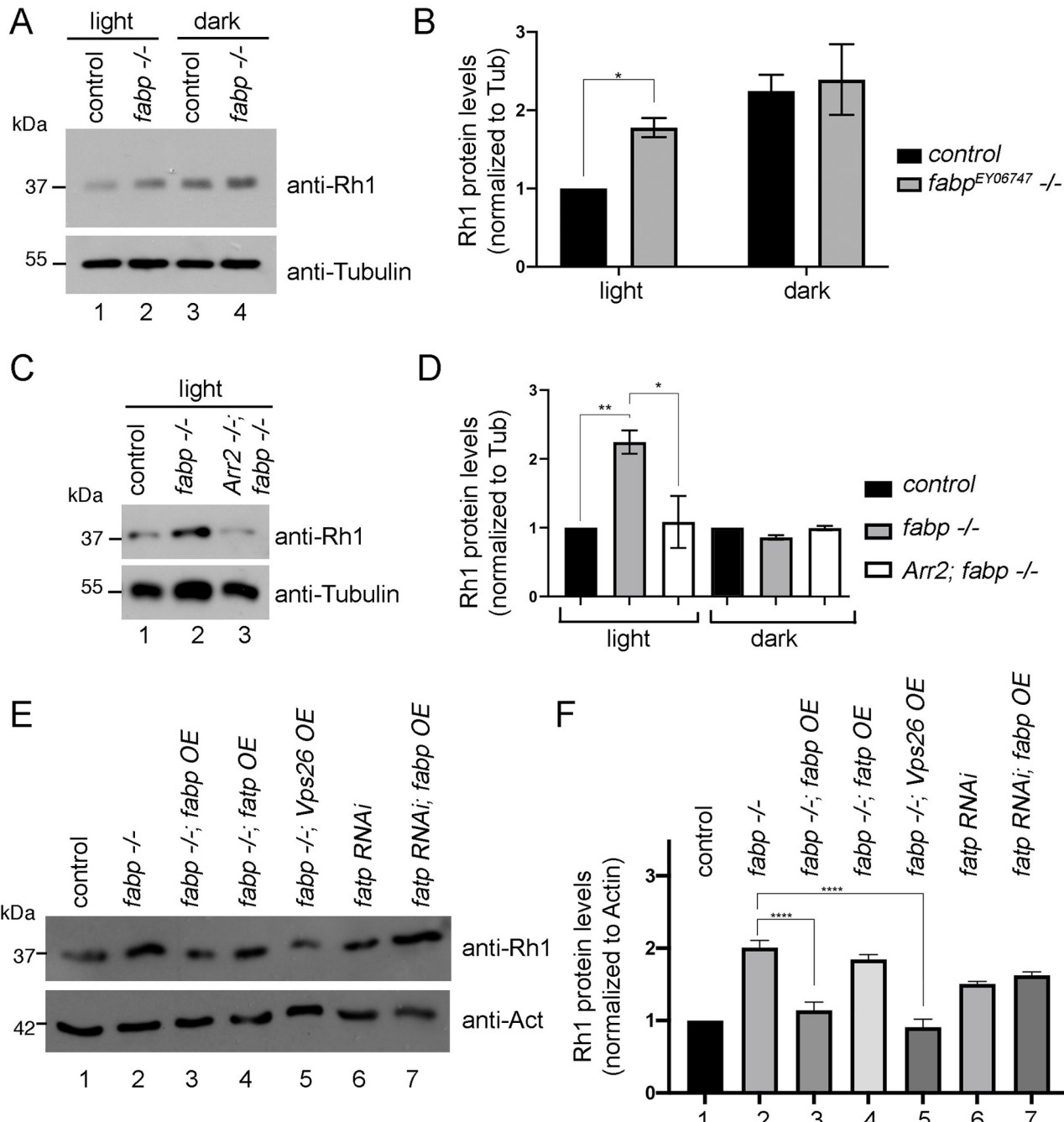

**Fig 5. *fabp* genetically interacts with *Arrestin2* and *Vps26* in regulating Rh1 levels in response to light exposure.** (A) An anti-Rh1 western blot of control *fabp wild type* (lanes 1, 3) or *fabp^{EY02678}* -/- (lanes 2, 4) fly head extracts. The flies were either reared under constant light (lanes 1, 2) or in constant darkness (lanes 3, 4). The lower band shows an anti-β-Tubulin blot as a control. (B) Quantification of anti-Rh1 blot intensities, normalized to anti-β-Tubulin blots. Statistical significance was assessed through a two tailed t-test. * = p<0.05. (C) Rh1 protein increase in *fabp* requires *Arrestin2*. Shown are western blots for anti-Rh1 (top gel) and anti-β-Tubulin (bottom) from fly head extracts of control *fabp wild type* (lane 1), *fabp^{EY02678}* (lane 2), *Arrestin2^3*; *fabp^{EY02678}* double mutant flies (lane 3). The flies were reared under constant light before being analyzed. (D) Quantification of the normalized anti-Rh1 blot intensities. Left three bars are from flies reared under constant light. The right three bars are results from those reared in constant darkness. Two tailed t-tests. * = p <0.05. ** p < 0.005. (E, F) Genetic interaction between *fabp* and *Vps26* in regulating Rh1 levels. Flies were reared under constant light prior to analysis. (E) Western blots of anti-Rh1 (top) and anti-Actin (bottom). *OE* indicates Over Expression of the indicated genes through the *GMR-Gal4* driver. RNAi knockdown was also performed using this *Gal4* driver. (F) *fabp^{EY06747}* -/- samples had higher Rh1 levels compared to controls (compare lanes 1 and 2). Overexpression of either *fabp* (lane 3) or *Vps26* (lane 5) in the *fabp^{EY06747}* -/- background suppressed such increase in Rh1 levels. *Fatp RNAi* also led to an increase in Rh1 levels (lane 6), but such effect was not suppressed by *fabp* overexpression (lane 7). Statistical significance was assessed through one way ANOVA with multiple comparisons test (compared to *fabp^{EY06747}* -/-). **** = p<0.00005.

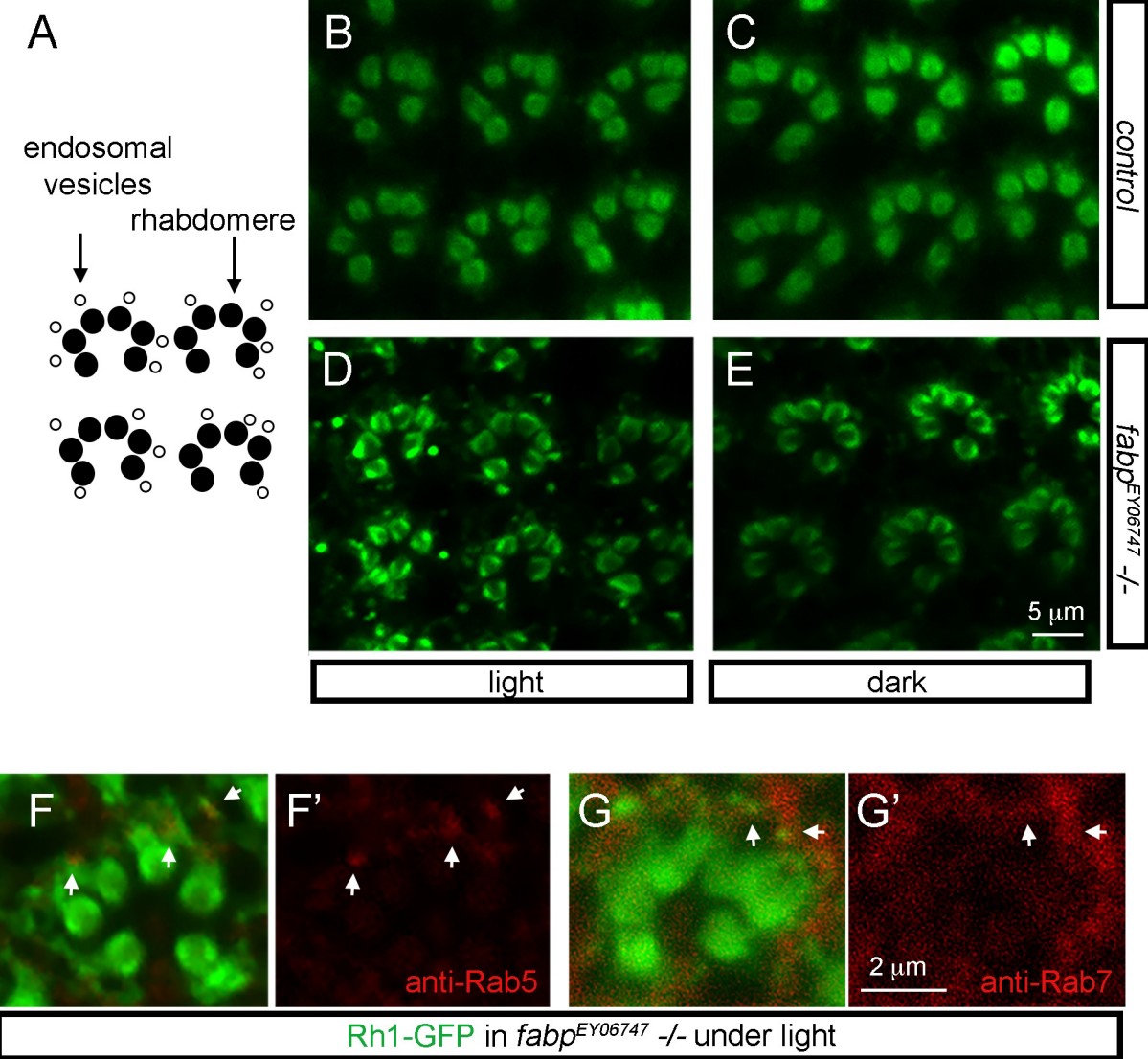

**Fig 6. Rh1 protein localization in _fabp^EY06747_ mutant eyes.** (A) A schematic diagram of adult _Drosophila_ ommatidia, with rhabdomeres labeled as black circles, and endosomal vesicles as white circles. (B-E) Anti-Rh1 labeling (green) of adult _Drosophila_ ommatidia. (B, C) Control _fabp wild type_ eyes. (D, E) _fabp^EY02678_ -/- eyes. Fly samples (B, D) were reared under constant light before being processed for fixation and immuno-labeling. By contrast, samples (C, E) were reared under constant darkness before processing. The scale bar in G applies for images D–G. (F, G) Assessment of Rh1 localization in _fabp^EY06747_ -/- flies reared under light through _ninaE-GFP_ expression (green). This line has GFP fused in frame with the Rh1 protein sequence, driven by the _ninaE_ promoter. Rh1-GFP is found in rhabdomeres and in cytoplasmic puncta. (F) Double labeling with anti-Rab5 antibody that marks early endosomes (red). (G) Double labeling with anti-Rab7 antibody that marks late endosome/lysosomes (red). Single channel images of the red channel are in (F', G'). White arrows point to representative regions of overlap. The scale bar in K' represents images in J and K.

We considered the possibility that increased Rh1 levels in _fabp^EY0674_ -/- is a contributing factor to retinal degeneration. To test this, we reduced the _ninaE_ gene dosage in _fabp_ mutants by introducing one copy of the _ninaE^17_ loss-of-function allele. TEM images of these flies at 27 days after eclosion did not show any signs of retinal degeneration (Fig 7C). The ommatidia were in regular arrays, each with seven visible rhabdomeres in trapezoidal patterns. These results indicated that reducing Rh1 levels suppress the retinal degeneration phenotype of _fabp^EY06747_ mutants.

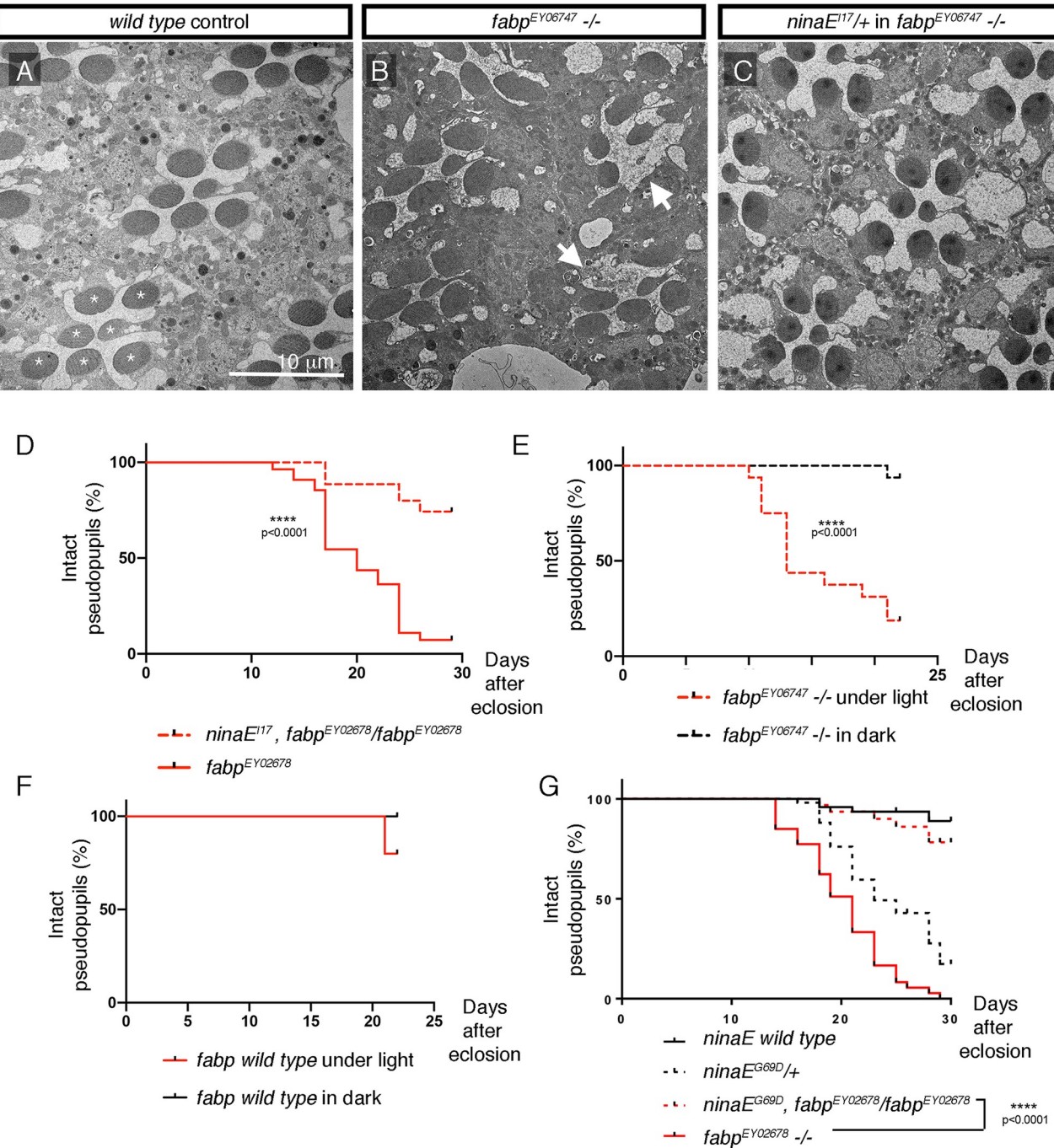

**Fig 7. Retinal degeneration in *fabp* loss of function mutants.** (A-C) Representative Transmission Electron Microscopy (TEM) images of adult fly ommatidia of the indicated genotypes. Flies were reared under 1000 lux constant light. (A) *ninaE wild type* control. Asterisks mark the seven rhabdomeres arranged in a trapezoidal pattern within a single ommatidium. Scale bar = 10 μm. (B) *fabp*$^{EY06747}$ -/- ommatidia from a 27 day-old fly raised under light. Two ommatidia on the right have less than seven rhabdomeres (white arrows), indicative of retinal degeneration. (C) Ommatidia of *fabp*$^{EY06747}$-/- flies in the *ninaE*$^{I17}$/+ background, also of 27 day-old flies, raised under light. (D, E) The course of retinal degeneration in flies of the indicated genotypes, assessed through Rh1-GFP pseudopupils. The y axis shows % of flies with intact photoreceptors. The x axis indicates the days after eclosion. (D) *fabp*$^{EY02678}$ mutant flies had accelerated retinal degeneration (solid red line), which was suppressed when one copy of *ninaE*$^{I17}$ loss-of-function mutant was introduced into the background (dotted red line, p < 0.0001, Log-rank test). (E) The course of retinal degeneration of *fabp*$^{EY06747}$ -/- when reared under 1000 lux of constant light (red dotted line), or reared under constant darkness (black dotted line). p<0.0001. (F) *fabp wild type* control flies reared under constant light (red solid line) or under constant darkness (black line). No statistical significance found (Log rank test). (G) The course of retinal degeneration in *fabp*$^{EY02678}$ mutant flies was also significantly delayed in the *ninaE*$^{G69D}$/+ background (p<0.0001, Log-rank test), even though *ninaE*$^{G69D}$/+ mutants showed age-related retinal degeneration phenotype on its own.

In order to validate these results in live flies, we used Rh1-GFP flies with their photorecep-tors labeled with green fluorescence. When hundreds of photoreceptors are in regular trape-zoidal array, they give rise to a single "pseudopupil" along the optical axis under low-power microscopy (S4 Fig). Under standard conditions in which the flies were exposed to moderate light, *fabp*$^{EY02678}$ -/- flies showed signs of abnormality as judged by the disappearance of such GFP-labeled pseudopupils: Specifically, a few flies of this genotype began showing the loss of Rh1-GFP pseudopupils around 12 days after eclosion, with almost all examined flies having signs of retinal degeneration by day 28 (Fig 7D, solid red line; also Fig 7E, dotted red line). When one copy of the *ninaE*$^{I17}$ loss-of-function allele was crossed into this background, the course of pseudopupil loss was significantly delayed, with a majority of the flies still maintain-ing pseudopupils at day 28 (Fig 7D, red dotted line; Log-rank test, p<0.0001). These results were consistent with the representative TEM image results, and further support the idea that excessive high Rh1 levels contribute to retinal degeneration in *fabp*$^{EY06747}$ mutants.

Retinal degeneration in *fabp* mutants was light-dependent, as those reared in the dark did not exhibit signs of photoreceptor degeneration (Fig 7E, black dotted line). The control *fabp* wild type flies showed no sign of pseudopupil loss up to 20 days after eclosion, regardless of light exposure status (Fig 7F). The light-dependent nature of photoreceptor degeneration cor-related with *fabp*'s effect on Rh1 levels.

We also used the GFP pseudopupil assay to examine *ninaE*$^{G69D}$/+ flies. These flies showed age-related retinal degeneration that started occurring around day 17, with most flies exhibit-ing retinal degeneration by day 30 (Fig 7G, black dotted line). Surprisingly, flies containing *ninaE*$^{G69D}$/+ in the *fabp* -/- background had a significantly delayed course of retinal degenera-tion, with most flies still showing intact Rh1-GFP pseudopupils 30 days after eclosion. While surprising, such genetic interaction with *ninaE*$^{G69D}$ is not unprecedented. Previous studies found that mutants that increase wild type Rh1 levels, such as *fatty acid transport protein* (*fatp*), cause severe retinal degeneration. Such retinal degeneration is suppressed in the *ninaE*$^{G69D}$/+ background [26]. We speculate that such suppressive effect could be due to the reduction of overall Rh1 levels in *ninaE*$^{G69D}$/+ eyes.

## Discussion

Photoreceptors tightly regulate light-activated Rhodopsin levels and a failure to do so could cause retinal degeneration. There are a number of reported genes involved in the degradation of light-activated Rh1 protein. Mutations in many of those genes result in endosomal accumu-lation of Rh1 in response to light, leading to retinal degeneration. Examples of this type include mutations in *norpA*, *culd*, retromer complex proteins, and *fatty acid transport protein* [18,24,26,45]. Here, we presented evidence supporting the role of *fabp* in regulating the endo-somal/lysosomal degradation of light-activated Rh1. Specifically, we found that Rh1 levels increase in *fabp* mutants when flies were reared under light, but not when the flies were reared in constant darkness. We conclude that the high level of Rh1 in the *fabp* mutant contributes to retinal degeneration, as conditions that reduce Rh1 levels suppress photoreceptor degenera-tion. Furthermore, Rh1 increase in *fabp* mutants were suppressed in the *Arr2* mutant back-ground. Immunohistochemical analysis shows that Rh1 accumulates in intracellular vesicles of *fabp* mutants only when the flies were reared under light. Since it is now well-documented that light-activated Rh1 undergoes endocytosis and lysosomal degradation [18–21], we interpret that *fabp* is specifically involved in this process.

Our photoreceptor-specific gene expression profiling analysis revealed that the ER-stress imposing *ninaE*$^{G69D}$/+ photoreceptors induce many genes involved in light-activated Rh1 clearance, which included *fabp*. This observation provides clues regarding a few previously

inexplicable $ninaE^{G69D}$/+ phenotypes: For example, it remained unclear why $ninaE^{G69D}$/+ causes a dramatic reduction of total Rh1 levels when the flies still have one wild type copy of the $ninaE$ allele. Our results with an epitope-tagged wild type Rh1 transgene now show that the wild type protein undergoes $fabp$-mediated clearance in $ninaE^{G69D}$/+ photoreceptors. Our results further indicate that $fabp$ contributes to retinal degeneration of $ninaE^{G69D}$/+ photoreceptors, as the age-related retinal degeneration phenotype of $ninaE^{G69D}$/+ was significantly delayed in the $fabp$ mutant background. These results provide new clues as to why retinal degeneration in $ninaE^{G69D}$/+ eyes are dependent on light exposure. The ER stress imposing property of $ninaE^{G69D}$ had been insufficient to explain such light sensitive nature, as there is no clear link between ER stress itself and light. The results presented here suggest that excessive degradation of light-activated wild type Rh1 protein by $fabp$ activity contributes to retinal degeneration in $ninaE^{G69D}$/+ photoreceptors.

$fabp$ initially drew our interest because of its homology to mammalian CRABPs, which are retinoic acid-binding proteins. We show that $fabp$ shares a few properties with CRABPs, including the effect of vitamin A and retinoids on $fabp$ expression. While this is an intriguing observation, there is as yet no evidence to support the existence of a retinoic acids signaling pathway in $Drosophila$ similar to those delineated in vertebrates. It is possible that vitamin A and retinoid dependent changes in $fabp$ expression occurs through a unique or indirect mechanism. Whether FABP binds to retinoids, and whether that property is necessary for Rh1 protein homeostasis is yet to be determined.

In conclusion, our study shows that $fabp$ is required for the homeostatic control of light-activated Rh1 proteins and for photoreceptor survival. Intriguingly, $fabp$ expression is induced in $ninaE^{G69D}$/+ flies that serve as a model for Retinitis Pigmentosa. It remains to be examined whether mammalian CRAPBs similarly regulate rhodopsin levels and affect retinal degeneration.

## Materials and methods

### Fly genetics

All fly crosses were maintained in 25˚C. Unless otherwise stated, flies were reared with a standard cornmeal-agar diet supplemented with molasses. Vitamin A deficient food was made by mixing 12 g yeast, 1.5 g agar, 7.5 g sucrose, 30 mg cholesterol, 3.75 ml of 1.15M Nippagin, 720 µl propionic acid in distilled water volume of 150 ml.

$Uas$-$fabp$ had EGFP fused in frame with the $fabp$'s N-terminal coding sequence. $EGFP$-$fabp$ was subcloned into the pUAST plasmid, and the resulting construct was injected by Best Gene, Inc., to generate the $uas$-$fabp$ transgenic line.

We used the following flies that had been reported previously: $Rh1$-$Gal4$ [52], $Rh1$-$GFP$ ($Rh1$ promoter driving GFP) [53], $ninaE$-$EGFP$ (GFP fused to the $Rh1$ coding sequence, driven by $Rh1$ promoter) [54], $ninaE^{G69D}$ [9], $santa\ maria^1$ [33], $ninaD^1$ [30], $Rh1$-$HSV$ [50], $uas$-$dicer2$ [55], $UAS$-$EGFP$::$Msp$-$300^{KASH}$ [42]. $fabp^{CA06960}$ [56] and $fabp^{EY02678}$ were obtained from the Bloomington $Drosophila$ Stock Center (stock numbers #50808 and #15579, respectively).

The RNAi lines used are as follows: $uas$-$lacZ\ RNAi$ [57], $uas$-$fabp\ RNAi$ (Bloomington Stock Center # 34685), $uas$-$fatp\ RNAi$ (Bloomington Stock Center # 55273), $uas$-$ninaB\ RNAi$ (Bloomington Stock Center #34994), $uas$-$knrl\ RNAi$ (Bloomington Stock Center # 36664), $uas$-$eg\ RNAi$ (Bloomington # 35234). These lines were crossed to the female virgins of the genotype: $Rh1$-$Gal4;\ ninaE^{G69D}$/TM6B. We collected non-TM6B progeny of these crosses to examine Rh1 protein and RNA.

## Photoreceptor-specific nuclear RNA extraction

We followed a published protocol to isolate Rh1-Gal4>UAS-EGFP::Msp-300$^{KASH}$-positive nuclei [42]. In brief, approximately 500 adult fly heads (from flies within 5 days of eclosion) per genotype were lysed in ice-cold nuclear isolation buffer (10 mM HEPES-KOH, pH 7.5; 2.5 mM MgCl$_2$; 10 mM KCl) with a dounce homogenizer. The homogenate was filtered through a 40μm Flowmi cell strainer (WVR, cat. #BAH136800040), and the filtrate was incubated with anti-EGFP-coupled protein G Dynabeads (Invitrogen, cat. #10003D) for 1 hour at 4˚C. The beads were collected using a magnetic microcentrifuge tube holder (Sigma, cat. #Z740155). Following washes with wash buffer (PBS, pH 7.4; 2.5mM MgCl$_2$), the beads were resuspended in a final volume of 150μL of wash buffer. Because the isolated nuclei remain intact until this stage, the ratio of the mRNAs within those nuclei should not be affected by any change in the levels of the Gal4 drivers. Then the post-isolation nuclei were suspended in 1mL of Trizol reagent (Life Technologies, cat. #15596018) for RNA extraction following standard procedures. Prior to RNA precipitation with isopropanol, 0.3M sodium acetate and glycogen were added to facilitate visualization of the RNA pellet. We then suspended the pellet in RNAse-free water and purified it using a Qiagen RNeasy MinElute cleanup kit (Qiagen, cat. #74204) following standard protocols.

## Preparation of cDNA libraries, RNA-seq and data processing

The NYU Genome Technology Center performed library preparation and RNA sequencing. We quantified RNA on an Agilent 2100 BioAnalyzer (Agilent, cat. #G2939BA). For cDNA library preparation and ribodepletion, we utilized a custom *Drosophila* Nugen Ovation Trio low-input library preparation kit (Tecan Genomics), using approximately 20 ng total RNA per sample. For sequencing, we performed paired-end 50bp sequencing of samples on an Illumina NovaSeq 6000 platform (Illumina, cat. #20012850) using half of a 100 cycle SP flow cell (Illumina, cat. #20027464). We used the bcl2fastq2 Conversion software (v2.20) to convert per-cycle BCL base call files outputted by the sequencing instrument (RTA v3.4.4) into the fastq format in order to generate per-read per-sample fastq files. For subsequent data processing steps, we used the Seq-N-Slide automated workflow developed by Igor Dolgalev (https://github.com/igordot/sns). For read mapping, we used the alignment program STAR (v2.6.1d) to map reads of each sample to the *Drosophila melanogaster* reference genome dm6, and for quality control we used the application Fastq Screen (v0.13.0) to check for contaminating sequences. We employed featureCounts (Subread package v1.6.3) to generate matrices of read counts for annotated genomic features. For differential gene statistical comparisons between groups of samples contrasted by genotype, we used the DESeq2 package (R v3.6.1) in the R statistical programming environment. We excluded genes with baseMean counts less than 300 so as to avoid artifacts due to varying extent of nuclei purification.

## Immunofluorescence and western blots

We followed standard protocols for western blots and whole mount immuno-labeling experiments using the following primary antibodies: Mouse monoclonal 4C5 anti-Rh1 (Developmental Studies Hybridoma Bank (DSHB), used at 1:5000 for western blots), anti-Rab7 (DSHB, used at 1:100 for immunohistochemistry), anti-Rab5 (Abcam #ab66746, used at 1:250 for immunohistochemistry), anti-Actin (Millipore Sigma #MAB1501, used at 1:2000 for westerns), anti-β tubulin (Covance #MMS-410P), Rabbit anti-GFP (Invitrogen #A-6455), anti-FABP antibodies [58].

## RT-PCR

We performed qRT-PCR using Power SYBR green master mix kit (Thermo Fisher). The primer sequences are as follows:

Rpl15F: AGGATGCACTTATGGCAAGC

Rpl15R: GCGCAATCCAATACGAGTTC

FatpF: CTCCCGGTGAGTGCAATAGCTT

FatpR: GCGGTGTGGTACAAAGGCAA

Arr1F: CATGAACAGGCGTGATTTTGTAG

Arr1R: TTCTGGCGCACGTACTCATC

Arr2F: TCGATGGAGTGATTGTGGTGG

Arr2R: GCGACCATAGCGATAGGTGG

Fabp-1F: CCGAGGTCTCAGTGTGCTC

Fabp-1R: CCGAGGTCTCAGTGTGCTC

Fabp-2F: CACAGTGGAGGTGACCTTGG

Fabp-2R: GATGCTCTTGACGTTGCGAC

TubF: CTCAGTGCTCGATGTTGTCC

TubR: GCCAAGGGAGTGTGTGAGTT

## Retinal degeneration assay

We performed all retinal degeneration assays in the *cn*, *bw* -/- background to eliminate eye pigments that otherwise affect the course of retinal degeneration. The flies were incubated in the 25˚C incubator with 1000 lux of light. For retinal degeneration assays under constant darkness, the flies were reared in an enclosed cardboard box in the 25˚C incubator. Retinal degeneration was assessed based on green fluorescent pseudopupils originating from the *Rh1-GFP* transgene. We interpreted clear trapezoidal pseudopupils as evidence in intact photoreceptors, while its disappearance was construed as a sign of retinal degeneration. The number of flies analyzed for each genotype in Fig 6E is as follows:

*wild type*, 48 flies; *fabp*$^{EY02678}$, 52 flies; *ninaE*$^{G69D}$/+, 50 flies; *ninaE*$^{G69D}$, *fabp*$^{EY02678}$/ *fabp*$^{EY02678}$, 32 flies.

For Fig 6F and 6G, 50 flies were analyzed for each genotype.

## Electron microscopy

Adult flies were anesthetized with $CO_2$ and heads were cut into half to ensure proper penetration of the fixative. The samples were put into freshly made fixative containing 2.5% glutaraldehyde, 2% paraformaldehyde and 0.05% Triton X-100 in 0.1M sodium cacodylate buffer (pH 7.2) on rotator for at least 4 hours until all fly eyes were sunk to the bottom of the tube, then change to same fixative without Triton and continue fixed at 4˚C for 4 days on rotator. After washing, the fly eyes were post fixed in 1% $OsO_4$ for 1.5 hour, dehydrated in a series of ethanol solutions (30%, 50%, 70%, 85%, 95%, 100%), followed by two rinses with propylene oxide and

embedded in EMbed812 epoxy resin (Electron Microscopy Sciences, Hatfield, PA). 500nm thick semi-thin sections were cut, mounted on glass slide and baked on hot plate overnight at 60°C. The sections are stained with 0.1% Toluidine blue, dried on hot plate and cover-slipped with Permount mounting medium (Electron Microscopy Sciences, Hatfield, PA) for light microscopy. 70nm ultra-thin sections were cut and mounted on formvar coated slot grids and stained with uranyl acetate and lead citrate. Imaging was performed by an electron microscope (CM12, FEI, Eindhoven, The Netherlands) at 120 kV, and recorded digitally using a camera system (Gatan 4k x 2.7K) with software Digital Micrograph (Gatan Inc., Pleasanton, CA).

## Quantification and statistics

To quantify proteins in gels, we measured average pixel intensities of western blot bands using Image J, and normalized them to anti-β tubulin or anti-Actin bands. Graphs were generated after at least three independent measurements and p values were calculated using paired t-tests. For Fig 5F that compared multiple genetic interactions, one way ANOVA test with multiple comparisons test was used. For retinal degeneration assays, we used the Log-rank (Mantel-Cox) text. Graphs were made using the *Graphpad Prism* program. All error bars represent SEM (Standard error of the mean).

## Supporting information

**S1 Fig. Knockdown of *fabp* enhances Rh1 levels in *ninaE^{G69D}*/+ flies.** (A, B) Western blot of Rhodopsin-1 (Rh1) and β-tubulin from fly heads extracts. (A) The first lane is from *ninaE wild type* samples. The remaining lanes are from *ninaE^{G69D}*/+ flies with the indicated genes knocked down through RNAi with the Rh1-Gal4 driver. *lacZ* RNAi (lane 2) was used as a negative control. (B) Quantification of relative Rh1 band intensities as normalized to β-tubulin. (JPG)

**S2 Fig. *fabp* regulates Rh1 levels in flies exposed to blue light, and genetically interacts with Arrestin2.** (A) Western blots of anti-Rh1 (top gel) and anti-β tubulin (bottom gel) in flies of the indicated genotypes that were exposed to blue light for 6 hours prior to analysis. (B) Quantification of Rh1 band intensities, normalized to that of β tubulin. The results indicate that *fabp* loss stabilizes Rh1 protein in flies exposed to blue light, and that function requires *Arrestin2*. Two tail t-tests were used to evaluate statistical significance. ** = p<0.005. **** = p<0.00005. (JPG)

**S3 Fig. Transmission Electron Microscopy (TEM) images of adult *Drosophila* ommatidia.** (A) Control fly eyes. Each ommatidium has seven rhabdomeres arranged in a trapezoidal pattern. These ommatidia are arranged in an array-like pattern throughout the adult eye. (B) Light-exposed *fabp^{EY06747}* -/- eyes at 27 days after eclosion. The ommatidial arrays show irregular patterns. Some ommatidia appear distorted, while others have missing rhabdomeres. There are vacuoles (marked with V) in between some ommatidia. (JPG)

**S4 Fig. Rh1-GFP pseudopupils visualized through low-power microscopy.** (A-D) Representative images of adult fly eyes of the indicated genotypes at 14 days after eclosion. White arrows point to the trapezoidal pattern of Rh1-GFP pseudopupils, which are indicative of intact photoreceptors. (JPG)

**S1 Table. Differential Gene Expression comparison between *ninaE wild type* and *ninaE^{G69D}*/+ photoreceptors.** Shown is a Table based on RNA-seq counts, generated through the DESeq2 package (R v3.6.1). The data are based on three replicate samples for each geno-type. The rows have been sorted by padj (adjusted p value). Positive log2FC (log2 value of Fold Change) indicates higher expression in *ninaE^{G69D}*/+ samples.
(XLSX)

**S2 Table. Source data for graphs.** The values of individual data points for the graphs displayed in the manuscript.
(XLSX)

## Acknowledgments

We thank Drs. Vikke Weake, Jason Gerstner for fly lines and antibodies, NYU Langone Genome Technology Center (RRID: SCR_017929) for assistance with RNA-seq, and NYULH DART Microscopy Laboratory (RRID: SCR_017934) Alice Liang, Chris Petzold and Kristen Dancel-Manning for assistance with electron microscopy work.

## Author Contributions

**Conceptualization:** Hyung Don Ryoo.

**Data curation:** Huai-Wei Huang.

**Formal analysis:** Huai-Wei Huang.

**Funding acquisition:** Hyung Don Ryoo.

**Investigation:** Huai-Wei Huang, Hyung Don Ryoo.

**Project administration:** Hyung Don Ryoo.

**Visualization:** Hyung Don Ryoo.

**Writing – original draft:** Hyung Don Ryoo.

**Writing – review & editing:** Huai-Wei Huang.

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
