## [Decision Letter · Decision Letter 0]

4 May 2021

Dear Dr Ryoo,

Thank you very much for submitting your Research Article entitled 'Drosophila fabp  is a retinoid-inducible gene required for Rhodopsin-1 homeostasis and photoreceptor survival' to PLOS Genetics.

The manuscript was fully evaluated at the editorial level and by independent peer reviewers. The reviewers appreciated the attention to an important problem, but raised some substantial concerns about the current manuscript. Based on the reviews, we will not be able to accept this version of the manuscript, but we would be willing to review a much-revised version. We cannot, of course, promise publication at that time.

If you decide to revise the manuscript for further consideration at PLOS Genetics, please aim to resubmit within the next 60 days, unless it will take extra time to address the concerns of the reviewers, in which case we would appreciate an expected resubmission date by email to plosgenetics@plos.org.

[LINK]

We are sorry that we cannot be more positive about your manuscript at this stage. Please do not hesitate to contact us if you have any concerns or questions.

Yours sincerely,

Bingwei Lu

Associate Editor

PLOS Genetics

Gregory P. Copenhaver

Editor-in-Chief

PLOS Genetics

Reviewer's Responses to Questions

**Comments to the Authors:**

Reviewer #1: Huang and Ryoo identified the fatty acid binding protein (fabp) gene was induced in ninaEG69D mutant fly eyes. Fabp is the fly ortholog of human CRABP-I, -II and FABP5. Among these human proteins, CRABP-I and -II have been shown to mediate retinoic acid signaling. Similar to CRABP-II, the authors showed that fabp expression could be induced by retinoid in cultured cells. They further demonstrated that a misfolding-prone rhodopsin-1 protein (ninaER69D) induced fabp expression, and loss of fabp alter photoreceptor functionality in vivo. This is an interesting study. Further mechanistic study, e.g. on the transcriptional regulation on fabp in the eyes and how fabp loss of function leads to photoreceptor malfunctioning should lead to a more thorough understanding of the role of fabp in Rhodopsin1 homeostasis.

Major comments:

Quality of data can further be improved.

Why only 3 DAPI signal were detected in Fig. 1C. What were the other GFP-positive signals?

It is not clear how the authors could differentiate properly folded Rhodopsin-1 protein over misfolded Rhodopsin-1.

Did the authors observe induction of fabp protein in S2 cells? Also, was Fabp protein induction observed when wildtype flies were treated with RA?

Had the ninaD experiment been performed (Figure 3) in fly eyes?

Did the authors observe induction of fabp-GFP genetrap expression in the eyes of ninaEG69D mutant?

Was RNAi gene knockdown and overexpression effect confirmed by RT-PCR and/or Western blot (Figure 4)?

Error bars are missing in Figure 6F and G.

Minor comments:

Quality of some figures can further be improved.

Quantification of eg, cg4372 and cg32821 is missing in Figure 4B.

Labeling is missing in Fig. 4D.

Figure 5: Endosomal vesicle marker should be included to demonstrate the identity of the green signals in Fig. 5F.

Molecular weight ladder and scale bar are missing in some blots/gels (Fig. 2A/C) and confocal micrographs (e.g. Fig. 1B and C) respectively.

Spelling mistakes are found in some figures (e.g. Figure 4E).

It is not clear which fabp mutant was used in Figure 5.

Line172 ninaE wild type flies: when describing wildtype flies, “ninaE” can be omitted.

Reviewer #2: In this manuscript, Huang and Ryoo identified fabp increased in ninaEG69D by photoreceptor-specific gene expression profiling. The authors found that fabp is a retinoid-induced gene, and functions in Rh1 hemeostasis and photoreceptor cell integrity. Moreover, loss of fabp could suppress the retinal degeneration of ninaEG69D. The author claimed that retinoid-inducible gene expression program might regulates Rh1 proteostasis and photoreceptor survival. Overall it is an interesting and important work. However, the manuscript lacks detail mechanisms. The authors showed that loss of fabp caused an accumulation of Rh1 proteins in cytoplasmic vesicles, but failed to give a convincing explanation. The authors showed that fabp is upregulated by ninaEG69D and/or retinoid acid, but did not provided by which pathway (for exmaple, ER stress) fabp is upregulated. It has been reported that mutations in fatty acid transport protein (fatp) caused adult-onset and progressive photoreceptor cell death through increasing Rh1 levels (PMID: 22844251), the phenotype is similar as fabp mutants, FABP is simple downstream of FATP or not.

Major points

It is important to check if FABP can bind to retinoid and which species retinoid it binds. If the retinoid levels and ratio is changed in fabp mutants. These biochemical characteristics might help to explain the genetic phenotypes of fabp mutants.

Authors first found that fabp transcription is induced by ninaEG69D, then they found that the it is inducible by Retinoic acid. The link between these two regulations is weak. The free retinoid levels might explain the phenotypes.

In mammals, retinoic acid serves as an important signaling molecule, but flies lack canonical retinoic acid signaling. The author found that fabp is regulated by retinoid acid, but there are not evidence supporting the transcriptional pathway that retinoid acid regulated in flies. Consider it is induced by misfolded Rh1, retinoic acid might induce ER stress response, which upregulates fabp.

Mutations in fatty acid transport protein (fatp) caused adult-onset and progressive photoreceptor cell death through increasing Rh1 levels (PMID: 22844251). This retinal degeneration is also suppressed by the ninaEG69D. It is important to check if FABP and FATP acts in same pathway and the epistasis of fabp and fatp in rhodopsin homeostasis.

Loss of increase fabp could increase rhodopsin levels in ninaEG69D/+ flies. It is assumed that the Rh1 that detected is wild-type form, which can explain the fact that degeneration can be rescued. It is important to have experimental evidence to support it.

Deep pseudopupil assay can reflect multiple defects, including Rh1 homeostasis, compactness of rhobdomere structure and cell viability. The retinal degeneration needs to be examined by TEM or other proper assays to support the conclusion.

The authors showed that fabp is retinoid-inducible, and found that it is requried for Rh1 homeostasis. however, if induction of fabp by retinoid is important for Rh1 is not convincing. Loss of fabp increased Rh1 levels in ninaEG69D and suppressed its retinal degeneration. As the retinoid regulated gene, it assumes that mutations disrupting retinoid metabolisms such as ninaD and santamaria downregulates fabp and have same effects as fabp mutations. However, in figure 4A, RNAi against ninaB seems did not affect the Rh1 in ninaEG69D

minor points

Authors used GFP trap lines to examine the response of FABP to retinoid in vivo. The major signals are detected in the gut, which is different with its endogenous function that authors proposed. The retina/photoreceptor cell signals should be checked.

Fig. 4D missing labels.

The MW markers should be indicated in Figs 2A, 2C, 4A, 4D, 4E and 5A.

Fig. 5 D-G need a quantification. As the Rh1 levels increased, the pictures that only showed increased vesicle fluorescence is not convincing. A endosomal marker need to be used if authors claimed that these are endosomal vesicle.

Reviewer #3: The study by Huang and Ryoo investigates retinoid-inducible genes in Drosophila, which have been poorly studied. They report a novel role for fabp a retinoid-inducible gene in Rh1 regulation and retinal degeneration. By performing a gene profiling, they first identified fabp, among others, as a gene which expression is increased in ninaEG69D/+ mutant compared to wild type retina. They next show that fabp levels are sensitive to retinoic acid treatments and mutant affecting retinoic acid metabolism. The loss of fabp function analysis revealed that Rh1 levels are upregulated in fabp mutant both in wild type and ninaEG69D mutants. Presumably fabp is required for lysosomal/endosomal Rh1 degradation in light dependent manner, shown by the accumulation of Rh1 vesicules that accumulate upon light in fabp mutants. They finally show that double mutant fabp and ninaEG69D have reduced degeneration compared to single mutants.

This is an original study which presents interesting and novel results. Most of the results are supported by data. I have some comments that may help to further improve the manuscript in particular about the mechanism regulating Rh1 upregulation in fabp mutant and the cause of retinal degeneration in fabp mutant. There is also a possible issue with the use of Rh1 driver in ninaEG69D mutant in Figure 1 and 6 that needs to be addressed.

-Regarding the method used for gene profiling I have an issue with the fact that Rh1-GAL4 was used to drive the expression of GFP:MSH in ninaEG69D, in which Rh1 transcription is upregulated. The authors need to comment on this and in particular if this could have introduced some kind of artifact, false positive or false negative.

Among the identified genes, the authors report few ER stress related genes but we could have expected others such as Hsc-3/Bip or xbp-1. Could it suggest that this approach missed some of the genes upregulated?

-Rh1 levels accumulate in fabp mutant retina presumably due to a reduced light-dependent degradation of Rh1. This part is the weakest and it would be nice to provide some additional analysis to support that claim. First the authors should show that Rh1 transcripts are not affected in fabp mutants.

Regarding light-induced degradation of Rh1, the authors should test if fabp mutants affect Rh1 degradation by blue light exposure. This would strongly support their claim that fabp is implicated in light-induced Rh1 degradation.

-Retinal degeneration in fabp is presumably due to the accumulation of Rh1 which when activated forms toxic complex with arrestin 2. The authors showed that degeneration is reduced in fabp ninaEG69D double mutant compared to single mutant which suggests that the reducing Rh1 level rescues fabp degeneration. Alternatively, fabp could reduce ER stress and protect from ninaEG69D-degeneration. To determine if indeed degeneration in fabp is due to Rh1 accumulation, the authors need to use the ninaEI17 null allele for the rescue. They should also perform a rescue for fabp degeneration in Vitamin A deficient food. Finally, an interesting experiment would be to determine if arrestin2 mutant rescues also fabp degeneration.

-There is a possible issue related with the upregulation of Rh1 transcription in ninaEG69D and the use of Rh1-GFP in the quantification of neurodegeneration in Figure 6. The upregulation of Rh1-GFP may have introduced a bias in this analysis. I recommend to repeat this quantification by using retina semi thin sections to visualize and count the rhabodmeres.

**Have all data underlying the figures and results presented in the manuscript been provided?**

Reviewer #1: Yes

Reviewer #2: Yes

Reviewer #3: Yes

PLOS authors have the option to publish the peer review history of their article (what does this mean?). If published, this will include your full peer review and any attached files.

Reviewer #1: No

Reviewer #2: No

Reviewer #3: **Yes: **Bertrand Mollereau

---

## [Decision Letter · Decision Letter 1]

20 Oct 2021

Dear Dr Ryoo,

We are pleased to inform you that your manuscript entitled "Drosophila fabp is required for light-dependent Rhodopsin-1 clearance and photoreceptor survival" has been editorially accepted for publication in PLOS Genetics. Congratulations!

Yours sincerely,

Bingwei Lu

Associate Editor

PLOS Genetics

Gregory P. Copenhaver

Editor-in-Chief

PLOS Genetics

Comments from the reviewers (if applicable):

Reviewer's Responses to Questions

**Comments to the Authors:**

Reviewer #1: The revised manuscript is suitable for publication.

Reviewer #2: The revised manuscript is significantly imporved by adding genetic interaction results between Arrestin2/Vps26 and fabp. My questions have been answered.

Reviewer #3: The authors addressed convincingly my comments with new data. The manuscript is significantly improved and easy to follow. I agree that at this stage the possible link with retinoids would require more work which is out of the scope of the current study. I am in favor for the publication of this article.

**Have all data underlying the figures and results presented in the manuscript been provided?**

Reviewer #1: Yes

Reviewer #2: Yes

Reviewer #3: Yes

PLOS authors have the option to publish the peer review history of their article (what does this mean?). If published, this will include your full peer review and any attached files.

Reviewer #1: No

Reviewer #2: No

Reviewer #3: **Yes: **Bertrand Mollereau

**Data Deposition**

http://datadryad.org/submit?journalID=pgenetics&manu=PGENETICS-D-21-00503R1

**Press Queries**

---

## [Editor Report · Acceptance letter]

26 Oct 2021

PGENETICS-D-21-00503R1 

Drosophila fabp is required for light-dependent Rhodopsin-1 clearance and photoreceptor survival 

Dear Dr Ryoo, 

We are pleased to inform you that your manuscript entitled "Drosophila fabp is required for light-dependent Rhodopsin-1 clearance and photoreceptor survival" has been formally accepted for publication in PLOS Genetics! Your manuscript is now with our production department and you will be notified of the publication date in due course.

With kind regards,

Zsofia Freund

PLOS Genetics

On behalf of:
